# Low-Velocity Impact Experiments and Modeling of TRC Skin-Aerated Concrete Core Sandwich Composites

**DOI:** 10.3390/ma14020390

**Published:** 2021-01-14

**Authors:** Chidchanok Pleesudjai, Anling Li, Vikram Dey, Barzin Mobasher

**Affiliations:** 1School of Sustainable Engineering & Built Environment, Arizona State University, Tempe, AZ 85287-3005, USA; cpleesud@asu.edu; 2Key Laboratory for Green & Advanced Civil Engineering Materials, College of Civil Engineering, Hunan University, Changsha 410082, China; all0629@hnu.edu.cn; 3Structural Project Manager, DCI Engineers, Los Angeles, CA 90017, USA; vikram.dey@asu.edu

**Keywords:** aerated concrete, textile reinforced concrete, sandwich composite, impact, digital image correlation, moment-curvature, flexural modeling

## Abstract

Mechanical response of textile-reinforced aerated concrete sandwich panels was investigated using instrumented three-point bending tests under quasi-static and low-velocity impact loads. Two types of core material were compared in the sandwich composite consisting of plain autoclaved aerated concrete (AAC) and fiber-reinforced aerated concrete (FRAC), and the stress skins were alkali-resistant glass (ARG) and textile reinforced concrete (TRC). The textile-reinforced layer promoted distributed cracking mechanisms and resulted in significant improvement in the flexural strength and ductility. Digital Image Correlation (DIC) was used to study the distributed cracking mechanism and obtain impact force-crack width response at different drop heights. A constitutive material model was also developed based on a multi-linear tension/compression strain hardening model for the stress-skin and an elastic, perfectly plastic compression model for the core. A detailed parametric study was used to address the effect of model parameters on the flexural response. The model was further applied to simulate the experimental flexural data from the static and impact tests on the plain aerated concrete and sandwich composite beams.

## 1. Introduction

Precast concrete sandwich panels have been used in the construction industry in North America for more than 60 years, since their thermal characteristics reduce the energy consumption of buildings due to heating and cooling [1]. A comparative evaluation of the embodied energy data of lightweight concrete with normal weight concrete, wood, or brick construction could result in 40–70% energy reduction during the entire life cycle [2]. Other positive attributes such as fire resistance, flexibility in shape, casting with precise dimension, and ability to span large vertical spaces between supports encourages the utilization of sandwich construction compared to other standard options such as wood, light-gauge steel, and exterior wall panels [1,3,4].

Aerated concrete (AC) uses a mixture of portland cement, high volume fly ash, quick lime, gypsum, water, and an expansive agent such as aluminum paste [3]. During the production process, a well-defined pore system is entrapped by the addition of metallic powders such as Al, Zn, or other gas forming agents [4]. The chemical reaction between aluminum paste and calcium hydroxide generates hydrogen gas, which forms a highly porous structure with a dry density of 400–800 kg/m^3^ and compressive strength values of 2–6 MPa for aerated concrete products [5]. Curing AC under pressure in an autoclave chamber prevents water movement and limits drying shrinkage [6]. Thermal conductivity and heat capacity describes the ability of heat to flow or be stored, respectively [7]. Thermal conductivity is in the range of 0.07–0.11 W/m°C, which is an order of magnitude lower than normal-weight concrete [8]. A mix design methodology to optimize thermal and mechanical properties by Yu et al. [9] showed that the structural efficiency defined as the ratio of compressive strength and density is not sufficient to compare different AC mixtures. This means that individual compressive strength or density are unable to indicate the thermal conductivity and efficiency. The experiments showed that finer lightweight aggregates (LWA) are more homogeneously distributed in the matrix than coarse LWA, which reduces the thermal conductivity. The properties of AC with high porosity, low permeability, and homogeneous aggregate distribution in the matrix are desirable [9,10]. AC products are primarily processed through an autoclaving industrial production to make Aerated Autoclaved Concrete (AAC). The hardened structure has a general ratio of 2.5:1.0 air-pores to micro-pores [11], which results in an 80% porosity discontinuous pore system.

Cellular structures exhibit considerable ductility and residual strength in post-peak compression [10]. The high ductility in compression is associated with pore crushing, but the ductility does not translate to enhanced flexural and tensile strength properties. Therefore, structural applications are limited to block construction [3]. Autoclaving provides higher strength due to the homogenization of the reaction products and reduces shrinkage potential when compared to moist curing [6].

In order to enhance the ductility in flexural and tensile response, about 0.5% by volume of short polypropylene fibers may be added to AC ingredients and normal curing used instead of autoclaving to make Fiber Reinforced Aerated Concrete (FRAC) [12]. FRAC exhibits lower compressive strength and higher variability than AAC, comparable thermal properties, and a flexural toughness as much as two orders of magnitude higher [3,13]. The short fibers affect the initiation, growth, and bridging of cracks due to mechanical, shrinkage, or impact stresses and are suitable core materials for sandwich composites [14]. The crack bridging of the fibers allows for the ductility based design using an elastic–quasi plastic approach for the core of sandwich composites [15]. Moreover, high ductility in the core can ensure full composite action with the shear transfer between two skin layers [16,17].

A sandwich panel with a ductile or brittle AAC core has potential applications for wall and roof elements by allowing higher flexural capacity, and longer span elements. Textile reinforced concrete (TRC) materials have been used for skin elements of sandwich composites by several researchers [16,18,19,20,21]. The skin is primarily involved in strength, stiffness, and load-carrying capacity, whereas the core acts as a thermal barrier and carries shear stresses. Utilization of TRC as a skin improves the panel surface characteristics and reduces the dead load, while its tensile strength and stiffness improves the overall system strength [19,22]. The high durability and aesthetic of external finishing [22,23] are also potential features that made TRC sandwich panels competitive.

Modeling of sandwich composites requires predictive tools for the design of structural elements [18,19]. Cuypers et al. [20] studied the behavior of sandwich panels with TRC faces using ANSYS finite element analysis [21]. TRC skin elements were used with IPC (inorganic phosphate cement) composites. The ACK model [24,25] was used for the behavior of the skin and the nonlinear behavior of the panels was compared with experimental observations. Djamai et. al. [26] studied the response of sandwich TRC composites using the ABAQUS finite element method. The textile–concrete bond was tested and modeled using the existing pull-out test results of the bond–slip law, which were then used to produce a multiscale numerical model based on matrix cracking and yarn pull-out damage parameters. Two approaches accounted for the mechanical behavior of a textile-reinforced concrete sandwich panel under a four-point bending test: a macroscopic finite element approach, which considers the composite TRC with a tri-linear stress–strain relation, as well as a multiscale 3D finite element approach, which uses a cohesive-friction bond–slip law evaluated as traction–separation law proposed by Turon et al. [27]. Wu and Wan [28] studied glass fiber reinforced polymer (GFRP) face sheets with foam core sandwich panels under low-velocity impact and compression after impact (CAI) tests. Fam et al. [29] developed a flexural model for sandwich panels of polyurethane foam core and glass fiber-reinforced polymer (GFRP) skins. Using equilibrium, strain compatibility, and both material and geometrical nonlinearity, their model incorporated various failure criteria of core shear, flexural tension and compression failure, compression skin crushing or wrinkling, and tensile rupture of skin. Nominal stresses at layer-by-layer were integrated over the cross-sectional area of the GFRP skins and the polyurethane foam core. These complex models necessitate the need for the development of closed-form solutions, since multiple phases of core and skin complicate the number of design variables, in terms of geometry and material parameters. In an area where numerical solutions of several parameters are the primary choice with so many interactive parameters and failure modes, the use of closed-form solutions for the design of sandwich panels is attractive.

In this paper, a closed-form solution model is developed using skin and core constitutive relationships to obtain the flexural load-deflection response of sandwich composites. Calibration of the model using experimental results allows for the extraction of material properties at various impact energy levels. The experiments consist of the flexural tests in quasi-static and low-velocity impact modes on sandwich composites with FRAC and AAC as core elements and TRC with alkali-resistant glass (ARG) textile as the skin element. The behavior of the beam specimens with different cross-sections and varying drop heights were tested and modes of failure were reported by Bonakdar et al. and Dey et al. [3,30]. The experimental results were evaluated using Digital Image Correlation (DIC) to further investigate the damage mechanisms, crack initiation, and propagation in the sandwich composites. By establishing crack length, location, and width measurements as a function of impact load, the modes of failure were studied. A multi-parameter constitutive model for the flexural moment–curvature relationship was developed using a series of interactive response zones constituting a combination of elastic, hardening, and softening for both tension and compression response of each of the phases. A permutation of failure mode interactions depending on the strain capacity and ductility of each phase is used to fit the experimental curves. This approach produced a predictive tool used as a basis of design and optimization for structural applications. Furthermore, a proper understanding of flexural response of sandwich panels helps the planning, design, and mobilization such as lifting and stacking to make sure that panels are able to maintain the performance characteristics during their service life.

## 2. Materials and Methods

### 2.1. Sample Preparation

AAC mixes were manufactured by using 18% Portland cement, 27% silica, 8% lime stone and gypsum, and 9% class F flyash. On the other hand, FRAC mixes was manufactured with 28% cement and 42% fly ash. Some 0.5% short PP fibers with a length of 12 mm were added to FRAC core in order to activate the ductility. Water content used in AAC was 38% and 28% in FRAC. It should be noted that mixture contents presented here are in percentage by weight. Static mechanical test results of the two core types are listed in Figure 1 tested by closed-loop displacement control [3]. The compressive strength values for the AAC and FRAC are 5.6 and 3.1 MPa, respectively, the higher strength of the AAC is attributed to the autoclave process, while FRAC was cured under room temperature. The toughness of FRAC at a displacement of 15 mm was reported to be 19 Nm [3]. This value is more than 60 times the toughness of AAC, and such a high difference is attributed to the effect of short polymeric fibers in FRAC.

Sandwich specimens were prepared in this study with AR glass TRC skins bonded to FRAC and AAC core panels. The skin layer of the sandwich composite used two layers of bonded AR glass textiles with a perpendicular set of warp and weft yarns glued at the junction points. In the absence of fibers in the core element, TRC-AAC can be characterized as a ductile skin-brittle core system, whereas TRC-FRAC can be considered as a ductile skin-ductile core system. Schematics of the assembly process of the different components of the sandwich members are shown in Figure 2a,b. The density of the textile in both warp and weft directions was four yarns per centimeter with 400 filaments of 13.5 μm in diameter. This resulted in an average fiber bundle of 0.27 mm in diameter with a tensile strength in the range of 1270–2450 MPa and modulus of elasticity about 78 MPa [31]. A blended cementitious paste with solids constituting 75% Portland cement (Type I/II) and 25% fly ash (class F) content was used as the binder for the TRC skin with a water/cementitious solids ratio of 0.15 and high range water reducing admixture of 0.2% by weight of the solids. During the casting of the sandwich specimens, equal portions of freshly mixed cement paste were placed layer-wise on the aerated concrete core with layers of AR-Glass textile placed in between them. Two layers of TRC were placed at the top and bottom face of the core sandwich composites were initially prepared with nominal dimensions of 300 × 300 mm in length and width using two different depths, 50 and 100 mm, as shown in Figure 2b. Using a mechanical press, compacting pressure was applied to the laminated surface to improve the interfacial bonding of samples before they were cured in a temperature and humidity-controlled chamber for 28 days at 23 °C with 95% RH. The beams were cut to the designated size by a water-cooled diamond blade saw with dimensions summarized in Table 1.

### 2.2. Static and Impact Tests

Three-point bending static tests were performed using a servo-hydraulic MTS under closed-loop conditions using the mid-span deflection control at the rate of 1.3 mm/min. The impact test set-up shown in Figure 3 is based on the free-fall of an instrumented hammer weighing 134 N on a specimen under three-point bending with a support span of 200 mm [32]. The hammer’s force is measured by strain-gage based load-cell of 90 kN capacity mounted behind the blunt shaped impact head. The initial force on the load cell was zeroed to account for the dead weight of the hammer. A linear variable differential transformer (LVDT) with a range of ±2.5 mm was connected to the tension zone utilizing a lever arm. A MATLAB code was developed for processing and interpretation of load, acceleration, and deflection. Time history data for the impact load, deflection, acceleration, and high-speed camera images were collected and analyzed.

### 2.3. Digital Image Correlation (DIC) Measurement

Digital Image Correlation (DIC) developed by Sutton et al. [33] and Bruck et al. [34] is an accurate non-contact optical deformation tool used in a wide range of stress analysis applications. DIC is used to measure the deformation and strain distributions in cementitious materials under static, high speed, and impact tests by various researchers [35,36]. A specimen with a speckled pattern is photographed using a high-resolution camera during its load-deformation impact event. A high-speed camera captures the impact event using a frequency of 3000 fps. A subset of pixels in the speckle pattern is used as an area of interest (AOI), and the position of the same pixels in the reference image and deformed images are tracked to calculate their corresponding translation using a cross-correlation function. This computation results in displacement and deformation; moreover, cracks are defined as discontinuities in the strain field, and their widths are measured by the software [37,38].

## 3. Results

### 3.1. Static Tests

Typical load-deflection trends from static tests and damage mechanisms of TRC-AAC and TRC-FRAC are shown in Figure 4a–d [13]. The load-deformation response shows three dominant zones starting with the linear–elastic behavior (Zone I). This stage is based on the iso-strain elastic response of the two phases of core and the skin and terminates by the formation of the first crack designated as the limit of proportionality (LOP). In Zone II, the nonlinear response is due to the formation of multiple cracks in both skin and core and terminates when the cracking is saturated. Finally, in Zone III, the response is characterized by damage localization, as the diagonal tension failure is observed in the core. There are two cases that may occur with brittle and ductile core materials. In TRC-AAC, brittle core, a shear dominated response, was observed as presented by softening deflection. Due to the presence of polypropylene fiber reinforcement in the Ductile TRC-FRAC, it was able to prevent premature shear failure in the core and exhibited strain hardening in zone III, as shown in Figure 4b, and tensile stiffening in skin face. The damage mechanism in these sandwich composites cannot be explained by a single mode of failure, but the dominant failure modes can be identified. The details of failure mechanisms are presented in the previous by Dey et al. [13] and the follow up sections.

The addition of TRC layers to the core significantly increased the toughness by a factor of 22 and 18 in TRC-AAC 50 × 50 and TRC-AAC 50 × 100, respectively. These measurements were compared to the plain AAC cores at deflection limit of 5 mm. The drastic increase in toughness in the case of TRC-AAC can be explained by the fact that the delamination of ARG stress skin was delayed due to multiple cracking in the core, while the use of fibers converted the brittle core to a ductile composite. Toughness of TRC-FRAC 50 × 50 and TRC-FRAC 50 × 100 increased by a factor 12 and 2, with respect to plain FRAC core [3]. It can be seen in Figure 4b that plain FRAC 50 × 50 shows deflection softening response after LOP, but FRAC 50 × 100 demonstrates a significant increase in the flexural load capacity and deflection hardening behavior, as observed in Figure 4d. The moderate toughness improvement in sandwich composites with FRAC core can be explained by the different failure mechanisms that are in effect. As mentioned earlier, shear failure is the governing failure mode for FRAC 50 × 50. It can be speculated that additional TRC skin layers in TRC-FRAC 50 × 50, which can prevent the composite from premature shear failure. The small span to depth ratio of FRAC 50 × 100 is enough to prevent shear failure and promotes crack bridging mechanism, thus enhancing energy absorption [14].

### 3.2. Comparison of Deflection Measurement with DIC Approach

Figure 5 shows the time–history curve of the midspan deflection of sandwich composite (TRC-AAC-B) specimen tested using a kinetic energy of 10.4 and 20.9 J corresponding to 75 and 150 mm drop height. Results from both the DIC analysis and the measurement by the LVDT are shown. The deflection signal by LVDT is lagging behind the impact force in the ascending response. There are also differences in the ascending and descending rates and maximum value between the deflections obtained by the LVDT and DIC methods. This is attributed to the frequency response of the conditioning system, which may be unable to capture the initial ascending rate and cracking; therefore, stiffness measurements may not be accurate from the LVDT signals.

Three main parameters explain the observed differences between the measurements in terms of the frequency response of time and amplitude lag by (a) high-frequency signals, (b) spurious deformations associated with the mechanical measurement of the displacement, and (c) relative vs. global nature of displacement measurement. The DIC based deformation is relative to the local deformation, while the LVDT deformation records the overall specimen deformation and the supports and thus includes the rigid body motion. The deflection obtained by the DIC method is considered more accurate than the LVDT and is therefore used for further analysis in Section 4.

The TRC skin increases the dynamic capacity several folds, as shown in Figure 6. The response of plain core is compared with sandwich composites at two depths of Type A (50 mm) and Type B (100 mm) with 150 mm drop height, respectively. The TRC layers facilitate the formation of multiple cracking and shear cracks and thus increase the load capacity due to the increases in section depth between the type A and B specimens. The response of the plain AAC core is quite brittle with little or no recorded deflection. However, sandwich composites show large deformations due to multiple cracking, delamination, and shear cracks. Size effect is observed between the two samples as the depth of the section is increased. Type A specimens show a steady state resistance at around 1500 N, whereas, in Type B samples, the load plateau increases up to 2500 N, representing an increase of 67% compared to the plain samples of similar size. The ductility of sandwich composites is at least two orders of magnitude higher.

### 3.3. Impact Process Analysis

Figure 7a shows the representative time history of force and deflection response of TRC-FRAC sandwich composite under impact loading with four stages of deformation. Stage I is the linear elastic range up to the limit of proportionality (LOP) represented by point A in Figure 7a. In this stage, the deformation increases slowly as the contact is made and then rapidly into a linear response. In stage II, multiple cracks form in the tension TRC skin, and the bridging effect of the microfiber in the core element results in the hardening response as the force increases to the ultimate point (B). Stage III is characterized by damage localization at point (C) as the deflection increases in the softening range due to propagation and widening of cracks. Stage IV is due to the lack of failure in the textile as rebound occurs by the release of elastic energy stored in the TRC skin as referred to by point D. Alternatively, one can observe for several specimens the formation of shear cracks and their propagation. This case occurs when the kinetic energy of the hammer sufficiently exceeds specimens ductility, resulting in a flexural crack through both phases. The post-impact stage V is characterized by total failure, and free vibration is observed.

### 3.4. Dynamic Response

The impact force-deflection response of sandwich composites for different heights is shown in Figure 8. The responses are similar for the TRC-ACC-B and TRC-FRAC-B specimens at each energy level. The dynamic flexural strength is calculated using the beam flexure equations [39] with the mean and standard deviation of three replicate samples presented in Table 2. Toughness values are calculated by the area under the impact force-deflection curve. Compared to TRC-AAC-B beams, the effect of drop height on the dynamic flexural strength and toughness is more significant for the TRC-FRAC-B beams, as the dynamic flexural strength and toughness increases from 2.3–3.9 MPa and 4.51–20.65 J with an increase in drop height from 75–300 mm, respectively. When the drop height is 300 mm, the dynamic flexural strength of the TRC-FRAC-B beam increases up to 3.9 (±0.7) MPa, which is 11.4% than TRC-AAC-B beams. Similarly, toughness for TRC-FRAC-B beams is 20.65 (±1.06) J, which is 7.2% greater than TRC-ACC-B beams.

Figure 9 shows the impact force-crack width response of sandwich composites. The high-speed camera captures the midspan zone of the specimen, which fails by the diagonal tension. Note that the shear crack width near the bottom of the specimen could not be obtained by the DIC, and the ultimate crack width may be larger than the reported values. For samples tested with the drop heights of 75 and 150 mm, the main flexural crack occurs near the center of the span with the maximum crack width less than 0.5 mm. With the drop height at 300 mm, shear cracks dominate the response, and the dominant crack is near the specimen support. The crack opening due to shear is sufficiently large due to the delamination of the TRC skin and core-skin debonding.

### 3.5. Failure Modes

The failure modes and cracking patterns for both types of sandwich composites at 300 mm drop height are compared in Figure 10. The shear failure mode dominates the sandwich response in high energy events with the diagonal tension cracks connecting the impact point and the supports. Local crush zones near the loading point and support as well as the interlaminar cracks between the core and skin are also observed. The tension TRC skin shows saturated lateral cracks and spalling regions. While there is debonding between the textile and matrix, rupture of textile yarns is not observed. Compared to TRC-ACC, the width and length of cracks of TRC-FRAC specimens are larger, and TRC-FRAC specimens show more flexural and shear cracks due to the bridging effect of the fibers in the core section of TRC-FRAC specimens. Failure modes are identified as fiber debonding and pullout, compression failure, delamination, flexural and shear cracks, cracks, and crushing at the support. These are labeled in Figure 10a,b with Roman numerals as specific modes of (I) compression failure and delamination, (II) intermediate crack, (III) shear crack, (IV) debonding, (V) flexural cracks, (VI) crushing at the support, (VII) delamination, (VIII) debonding and pullout.

## 4. Flexural Modeling of Sandwich Response

### 4.1. Idealized Constitutive Relations of Core and Skin and Stress Distributions

The multi-linear parameterized tension and compression models for TRC and AC are shown in Figure 11. The properties are defined by a single term, the first crack tensile strength of the core as $\sigma_{cr}=\varepsilon_{cr}E$, that is defined by two intrinsic parameters of the first crack tensile strain and elastic Young’s modulus ($\varepsilon_{cr},E$) as shown in Figure 11a. To use a minimum number of variables, every other parameter is defined as normalized non-dimensional variables through a series of control points defined as following: the applied strain at the tensile fiber interface is defined as the only independent variable in the model represented as *β* defined as $\varepsilon =\beta \;\varepsilon_{cr}$ such that elastic range is defined as 0≤β≤1 and post cracking as β>1. After cracking in the tensile domain, a constant residual strength defined by $\sigma_{ult}=\mu \sigma_{cr}$ is used, with μ < 1 representing the non-dimensional residual tensile strength. The limit of tensile strains is defined with the variable $\varepsilon_{ult}=\beta_{ult}\varepsilon_{cr}$ where $1\leq \beta_{ult}\leq \infty$. The following non-dimensional parameters with the exception of tensile core strength ($\sigma_{cr},\varepsilon_{cr},E$) will be defined as: the first subscript of each parameter corresponds to the type of the stress (tensile (t) or compressive (c)), the second subscript to the domain (core (c), or skin (s)), and the third subscript to the stress/strain regime (1, 2 or ultimate(ult)).

The core compression stress–strain is defined through the normalized variables for stiffness and compressive strength of core as $\gamma_{cc}$ and $\omega_{cc,1}$ and defined as $E_{cc}=\gamma_{cc}E$, $\varepsilon_{cc,1}=\omega_{cc,1}\varepsilon_{cr}$, and $\sigma_{cc,ult}=\omega_{cc,1}\gamma_{cc}\sigma_{cr}$, as shown in Figure 11b. A similar approach is adopted for representing the parametric tension and compression properties of skin sections. It is assumed that skin and core are perfectly bonded and no delamination is present with a uniform strain at the interface of skin and core. Furthermore, it is assumed that due to the small thickness, the state of stress in the skin is uniform and constant. The skin parameters are defined as multilinear strain hardening for both tension and compression, as shown in Figure 11c,d. Tensile stiffness and strains are defined as follows. First cracking strain in tension is defined as $\varepsilon_{ts,1}=\beta_{ts,1}\varepsilon_{cr}$, followed by the characteristic damage state [40] defined at strain $\varepsilon_{ts,2}=\beta_{ts,2}\varepsilon_{cr}$ and terminated at $\varepsilon_{ts,ult}=\beta_{ts,ult}\varepsilon_{cr}$. Compressive skin strain is $\varepsilon_{cs,1}=\omega_{cs}{}_{,1}\varepsilon_{cr}$ and terminated at $\varepsilon_{cs,ult}=\omega_{cs,ult}\varepsilon_{cr}$. The skin elastic modulus is $E_{ts}=\gamma_{ts}E$ and $E_{cs}=\gamma_{cs}E$, and the reduction in the skin stiffness after first cracking is $\eta_{ts,1}$ and $\eta_{ts,2}$ for tension and $\eta_{cs,1}$ for compression, which is positive for strain hardening and negative for strain softening.

The normalized stress–strain model for core and skin are listed next. Core materials compression and tension constitutive response are defined as a function of normalized strain, *β*.
(1)$$\frac{\sigma_{cc}\left(\beta \right)}{E\varepsilon_{cr}}=\left\{ \begin{array}{l} \gamma_{cc}\beta \;\;\;\;\;\;\;\;0\leq \beta \leq \omega_{cc,1}\\ \gamma_{cc}\omega_{cc,1}\;\;\;\;\;\;\;\;\omega_{cc,1}<\beta \leq \omega_{ult} \end{array}\right.\;\;\frac{\sigma_{tc}\left(\beta \right)}{E\varepsilon_{cr}}=\left\{ \begin{array}{l} \beta \;\;\;\;\;\;\;0\leq \beta \leq 1\;\;\;\;(k=\frac{1}{2})\;\\ \mu \;\;\;\;\;\;\;1<\beta \leq \beta_{tc,ult}\; \end{array}\right.$$

Skin material’s compression and tension constitutive responses are:(2)$$\begin{array}{l} \frac{\sigma_{cs}\left(\beta \right)}{E\varepsilon_{cr}}=\left\{ \begin{array}{l} \gamma_{cs}\beta \;\;\;\;\;\;\;\;\;\;\;\;\;\;\;\;\;\;\;0\leq \beta \leq \omega_{cs,1}\\ \omega_{cs,1}\gamma_{cs}+\eta_{cs,1}\gamma_{cs}(\beta -\omega_{cs,1})\;\;\;\;\;\;\;\;\;\;\;\;\omega_{cs,1}<\beta \leq \omega_{cs}{}_{,ult}\; \end{array}\right.\\ \frac{\sigma_{ts}\left(\beta \right)}{E\varepsilon_{cr}}=\left\{ \begin{array}{l} \gamma_{ts}\beta \;\;\;\;\;\;\;\;\;\;\;\;\;\;\;\;\;\;\;0\leq \beta \leq \beta_{ts,1}\\ \beta_{ts,1}\gamma_{ts}+\eta_{ts,1}\gamma_{ts}\left(\beta -\beta_{ts,1}\right)\;\;\;\;\;\;\;\;\;\;\;\beta_{ts,1}<\beta \leq \beta_{ts,2}\\ \beta_{ts,1}\gamma_{ts}+\eta_{ts,1}\gamma_{ts}\left(\beta_{ts,2}\;-\beta_{ts,1}\;\right)+\eta_{ts,2}\gamma_{ts}\left(\beta -\beta_{ts,2}\right)\;\;\beta_{ts,2}<\beta \leq \beta_{ts,ult} \end{array}\right. \end{array}$$

The cross-sectional geometry of the sandwich is shown in Figure 12a with the dimensions of the core as “*b”*, and “*h*” and skin thickness parameterized using variable “*ρ*“ as “*ρh”*. The composite interface between skin and core is assumed to be sufficiently strong in shear [16,17]. Assuming that the section has a linearized strain distribution along with the depth as shown in Figure 12b, the location of the neutral axis is *“kh”,* where *k* represents a non-dimensional depth. Piecewise linear properties of core and skin materials as elastic-hardening or perfectly plastic [41] in both tension and compression, as shown in Figure 11, are used. The internal force and moment can be computed as a function of *k* using the integration of forces and couples in non-dimensional terms:(3)$$F=\int_{A}\sigma dA=b\int_{-h/2}^{h/2}\sigma (y)dy\;\;\;\;\;M=\int_{A}\sigma ydA=b\int_{-h/2}^{h/2}\sigma (y)ydy$$

By expressing stress as piecewise linear, the force and moment components are obtained as quadratic and cubic functions, respectively. Location of the neutral axis, *k*, is obtained by solving the quadratic equation of equilibrium of internal forces as earlier work and used in the calculation of the resultant moment.

### 4.2. Moment–Curvature Response of the Sandwich Sections

The sandwich composite response is based on the perfect bonding between the isotropic core and skin. To generate the moment–curvature relationship in closed form, the equilibrium, compatibility, and constitutive laws are integrated. The tensile and compressive strain distributions in core and skin are expressed in multi-linear terms of maximum tensile strain β as shown in Figure 13 and integrated across the depth to obtain the equilibrium of internal force. Integration of piecewise linear stress terms involves the computation of areas of triangles, rectangles, and trapezoidal segments and results in closed-form solutions to quadratic equations for the position of neutral axis for any given tensile strain β [42]. The curvature and moment obtained for each strain measure, β is:(4)$$\phi =\frac{\beta \varepsilon_{cr}}{(1-k)h}$$

The combination of various modes of stress distribution depends on the input parameters that define the linear segments. The $M_{i}$ and $\phi_{i}$ at each stage are obtained for each strain $0<\beta \leq \beta_{tu}$ and expressed by the normalized parameters $m\prime_{i}$, and $\phi \prime_{i}$ and cracking terms $M_{cr}$ and $\phi_{cr}$. For each mode imposed, a different moment–curvature response is obtained, and the interaction of these responses from different zones are controlled by the variable limits as shown in Figure 14. Full derivations for the moment, curvature and neutral axis $M_{i}$, $\phi_{i}$, and $k_{i}$ as defined in Equation (5) as a function of the material properties are presented in Pleesudjai [42].
(5)$$M_{i}=m\prime_{i}M_{cr};\;\mathrm{and}\;\phi_{i}=\phi \prime_{i}\phi_{cr}\;M_{cr}=\frac{1}{6}bh^{2}E\varepsilon_{cr};\;\mathrm{and}\;\phi_{cr}=\frac{2\varepsilon_{cr}}{h_{0}};$$

Two available approaches for calculation of the load-deflection response include moment–area method and a closed-form solution using a simplified linearized tri-linear model [43,44,45]. The first approach was used by integrating the moment–curvature distribution over the length of the beam to obtain the rotation and deflection. Multiple cracking requires one to use localization zone, *L_p_*, in the flexural deflection calculations and the specimen depth was used for this parameter [36].

### 4.3. Parameter Estimation From Experimental Data

Test results on the tensile and compressive stress–strain response of AAC and FRAC from Bonakdar et al. [3] were used. The constitutive responses were linearized and shown in Figure 15a,b, respectively. TRC skin material properties were obtained from [27,34], as illustrated in Figure 16. The ratio of compressive to tensile strength is quite large to the order of $\gamma_{cc}\omega_{cc,1}$ = 28–52; hence, the response in flexure is dominated by tensile strength and shear. For the compressive response, a simplified bilinear model tends to fit the ultimate strength, and the plastic range can be overestimated, since the compression ductility is irrelevant, since even if it is treated as a brittle failure and pore crushing ignored, little difference in the simulation will be observed. AAC is treated as a brittle material with a similar elastic modulus as FRAC in tension, and residual tensile strength *μ* = 0 for AAC while *μ* = 1 is used for FRAC. The flexural failure is by the ultimate tensile strength of skin and core and occurs before reaching a plastic compressive response. The normalized parameters for the reference study are listed in Table 3 for core and skin elements. The shear effects are however not considered since the formulation would be different [46,47].

### 4.4. Model Validation with Experimental Sandwich Composites

Static flexural experimental data obtained from Dey et al. [30] were simulated using the proposed model. The original testing program was focused on evaluating two primary studies, the effect of core materials, TRC-FRAC vs TRC-AAC, as well as the effect of section depth, 50 and 100 mm. To validate the numerical model, both TRC-FRAC and TRC-AAC are simulated for both dimensions, and the efficiency of the sandwich enhancements are compared with the plain core flexural specimens. Both sets of results are shown in Figure 17 and Figure 18. When the sandwich composite is considered, the material response changes significantly with the change in thickness. The skin materials have a much higher stiffness than the core by definition of parameters *γ**_ts_* and *γ**_cs_*, and their ratio was around 17, as reported in Table 3. Hence, the skin carries a major part of the internal force while the core resists the shear.

The effect of skin is shown after stage B in TRC-AAC (see Figure 17) and beyond the flexural load of 500 N in TRC-FRAC (see Figure 18). The load-deflection response matches the initial deflection region when the flexural cracks are dominant. For TRC-AAC, after flexural load passes stage B, the core loses capacity. At this stage, the load transfer from the composite to the skin layers is shown by gradual reduction of the load, while shear crack propagation appears in DIC between stages B and C as shown in Figure 17. From stage C to stage D, the specimen maintains equilibrium with increasing load as the skin is the primary carrier, and the bond results in successive stable cracks in the core. Additional load results in cracks that coalesce, and at the stage from D to E, a diagonal tension crack travels through the core depth leading to failure. TRC-FRAC specimens show a better fit before point C, with dominant flexure deformations. The FRAC core seems to control diagonal tension cracks better using the residual tensile strength as shown in Figure 15a.

Crack distribution depends on the magnitude of tensile and bond stresses, as multiple cracking in core materials demonstrate the function of the skin in distributing the forces and nonuniform strain through the tensile layer. The crack width reduction due to the reinforcing fibers in the core contributes to a more homogeneous strain distribution [36]. The characteristic length for the average curvature in the localized zone is crack spacing multiplied by the number of cracks *L_p_* = *nS* [48]. Gradual crack formation and stiffness reduction can be replicated using parameters for sequential crack formation.

Figure 18 and Figure 19 show the relationship between the number of cracks and the stiffness reduction in TRC-FRAC specimens. As multiple cracks form, the gradual stiffness reduction is documented by successive curves, and the load-deflection response follows the experimental data. The deviation of the flexural cracking model and the experiments points to the ultimate failure by the diagonal tension cracking. The shear response can be modeled by similar approaches; however, it is beyond the scope of the present work.

### 4.5. Simulation of Flexural Impact Response

The present model can be used to address the response of sandwich composites subjected to impact by changing the constitutive properties. The strain rate in the core and skin is assumed to experience the same general response. However, major flexural cracks along the depth were followed up to the point where delamination was observed; at that point, due to the lack of shear strength failure criteria, the model is unable to switch over and predict the shear failure.

Simulation of the response of TRC-AAC and TRC-FRAC under different hammer drop heights is presented in Figure 20 and Figure 21, respectively. The strain-rate sensitivity of individual materials and their fracture energy in TRC-FRAC is higher than the TRC-AAC, which is less strain-rate sensitive. One can define a dynamic increase factor (DIF) on both unreinforced and reinforced matrices [49,50]. Additional simulations are needed to address the strain-rate sensitivity of bond-strength between fiber-matrix and skin-core systems. For DIF of plain concrete strength, it can be observed that tensile properties are the most sensitive to the impact load. As a result, this simulation attempts to address the tensile parameters, while keeping compressive and skin’s material properties in the same range as static stimulation in Table 3.

To simulate impact response, components of piecewise linear variables were estimated using the inverse analysis approach. Young’s modulus in both sandwich composite present higher than the static test, ranging from 148 to 202 MPa, and cracking strain was observed two times higher, at 2080 με. For TRAC-FRAC, residual tensile strength remains within a similar range of the static model at around 0.12 MPa. Moreover, the results show that peak tensile strength in TRC-AAC slightly decreases by 9% with increased drop height, whereas the peak tensile strength in TRC-FRAC increases by 9% and 17%, respectively. However, the most sensitive factor in the simulation model is observed in the tensile toughness as listed in Table 4. The tensile toughness in TRC-FRAC is around 230–500% of TRC-AAC at the corresponding drop height.

## 5. Conclusions

Flexural experiments on textile-reinforced aerated concrete sandwich panels were conducted under quasi-static and low-velocity impact loads. Plain and fiber-reinforced aerated concrete were used as core and AR-glass textile concrete was used as the skin element. Specimen responses were characterized as ductile skin with ductile and brittle cores. A parametric model was proposed to simulate the contribution of the core and skin phases. Closed-form moment–curvature relationship was obtained by the integration of section equilibrium, strain compatibility, and piecewise linear constitutive law. The interaction of the material properties in terms of strain capacity and stiffness of each phase were addressed by a parametric model for the individual core and skin. The model was used for the back-calculation of the tensile response in the core as a function of the drop height. This approach allows parameter estimation as well as a standard procedure to analyze and design sandwich composites. The key experimental findings are summarized below:The addition of TRC layers significantly increased the toughness of AAC sandwich panels by a factor of 22 and 18 for specimen depths of 50 and 100 mm, respectively, as compared to plain AAC. At deflection limit of 5 mm, the toughness of TRC-FRAC increased by a factor 12 and 2, for specimen depths of 50 and 100 mm, respectively, when compared to plain FRAC specimens.In high energy events, the shear failure mode dominated the sandwich response, with the diagonal tension cracks defining the ultimate strength point. The tension TRC skin carries a significant amount of load and provides the strength and ductility to the composite. The damage mechanism of the sandwich composite is characterized by saturated lateral cracks and localized spalling in the TRC skin, as well as debonding between the textile and matrix layers within the skin.Increase in drop height from 75–300 mm correlated with an increase in the dynamic flexural strength and toughness, which increased from of 2–8 MPa and 4–21 J for TRC-FRAC, respectively. Whereas for the similar range of drop height the increase in flexural strength and toughness for TRC-AAC increase from 2–9 MPa and 3–26 J, respectively.The simulation of flexural response by the proposed model correlated with the DIC deflection measurements. The difference between the LVDT and DIC was due to three main parameters: (a) frequency response of the signals, (b) spurious deformation of the mechanical measurement of the displacement, and (c) differential displacement by DIC vs. global displacement by LVDT.The interaction of the material properties in terms of strain capacity and stiffness of each phase can be addressed by the parametric model and used for the optimization of the composite property.Simulation of the flexural response depends on the length of localization zone, which is used to match the sequence of cracking obtained from the DIC analysis.The shear failure criteria is an important aspect of the core failure mode.

## Figures and Tables

**Figure 1 materials-14-00390-f001:**
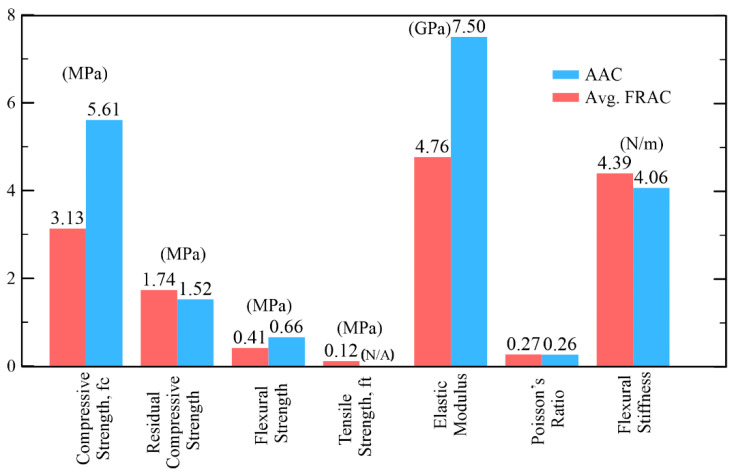
Selected materials properties for aerated concrete [3].

**Figure 2 materials-14-00390-f002:**
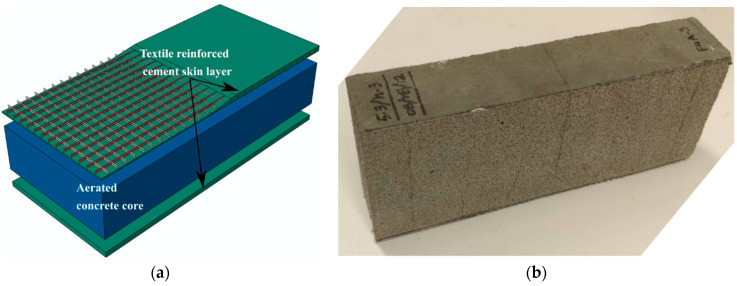
(**a**) Schematic of the textile reinforced-aerated concrete sandwich composite system, (**b**) sandwich beam specimen.

**Figure 3 materials-14-00390-f003:**
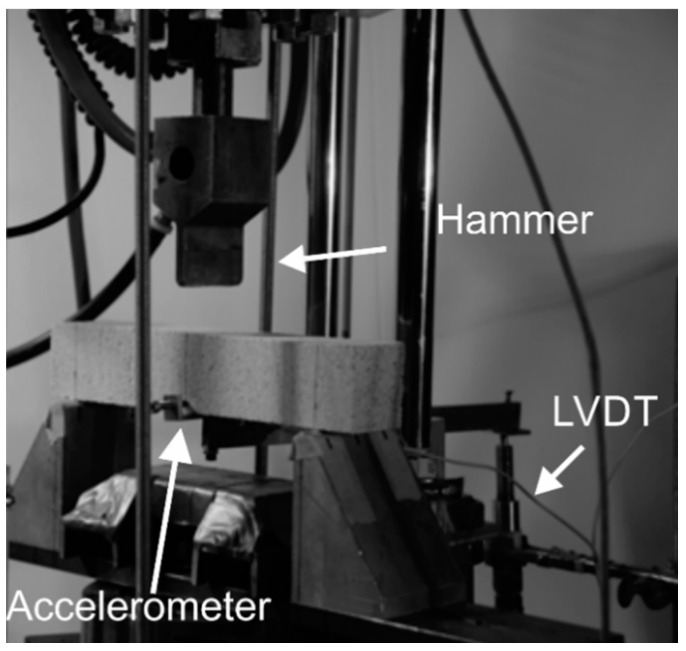
Impact test set-up.

**Figure 4 materials-14-00390-f004:**
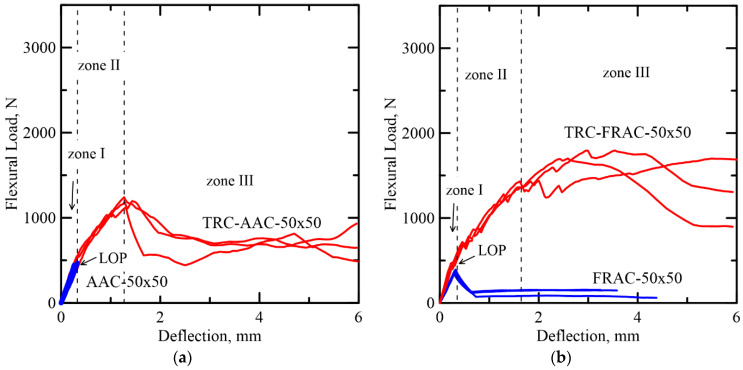
Comparison of the static test results of: Category A samples. (**a**) AAC and (**b**) FRAC and category B samples. (**c**) AAC and (**d**) FRAC.

**Figure 5 materials-14-00390-f005:**
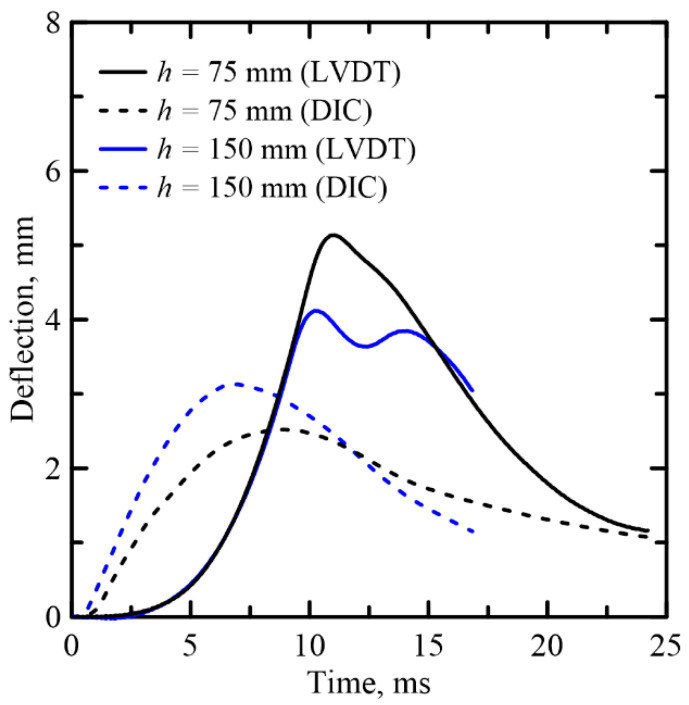
Comparison of deflection-time curves obtained by LVDT and DIC for TRC–FRAC-B series.

**Figure 6 materials-14-00390-f006:**
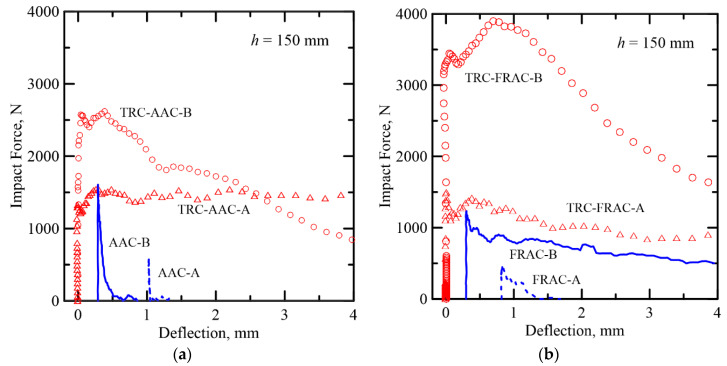
Comparison of (**a**) plain core and sandwich AAC composites, and (**b**) plain core and sandwich FRAC composites for two different specimen sizes (x-abscissa has been shifted to reflect the differences).

**Figure 7 materials-14-00390-f007:**
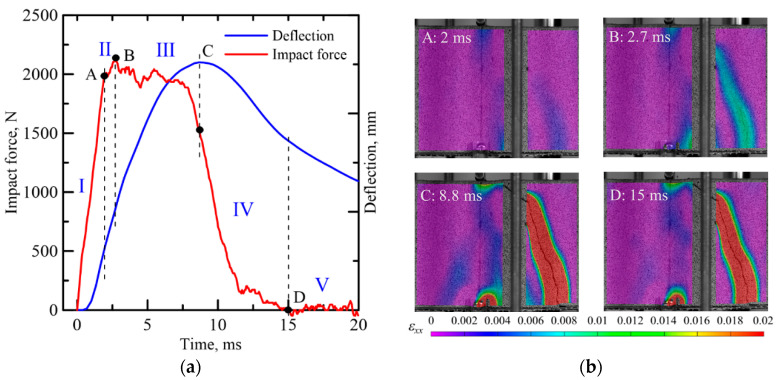
(**a**) Time history curves of impact force and deflection for TRC-FRAC composite under impact using DIC. (**b**) The specimen surface Lagrangian strain by VIC-3D of Zones I–II through V.

**Figure 8 materials-14-00390-f008:**
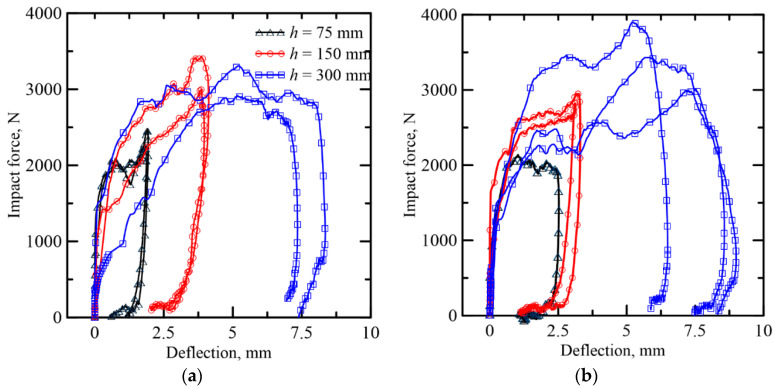
Dynamic response of sandwich composite: (**a**) TRC-ACC (B series), (**b**) TRC-FRAC (B series).

**Figure 9 materials-14-00390-f009:**
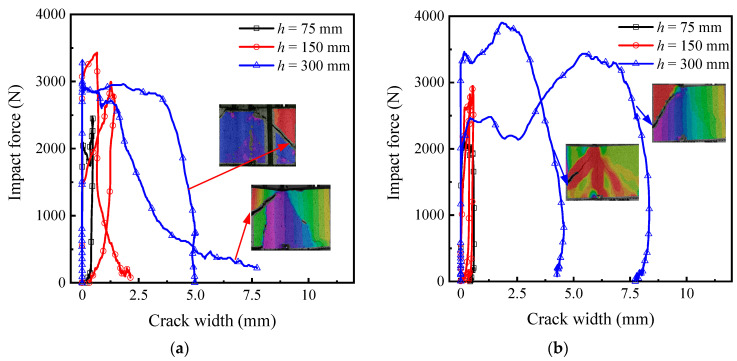
Impact force-crack width of sandwich composites (**a**) TRC-ACC(B series), (**b**) TRC-FRAC (B series).

**Figure 10 materials-14-00390-f010:**
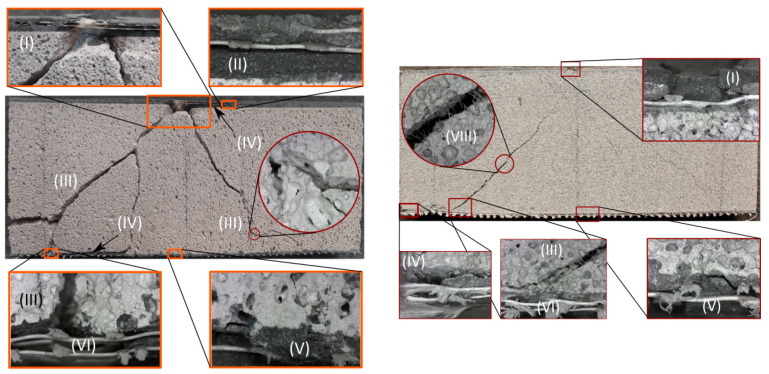
Failure modes and cracking patterns of sandwich composites under impact loading with 300 mm drop height, (**a**) TRC-ACC-B, (**b**) TRC-FRAC-B.

**Figure 11 materials-14-00390-f011:**
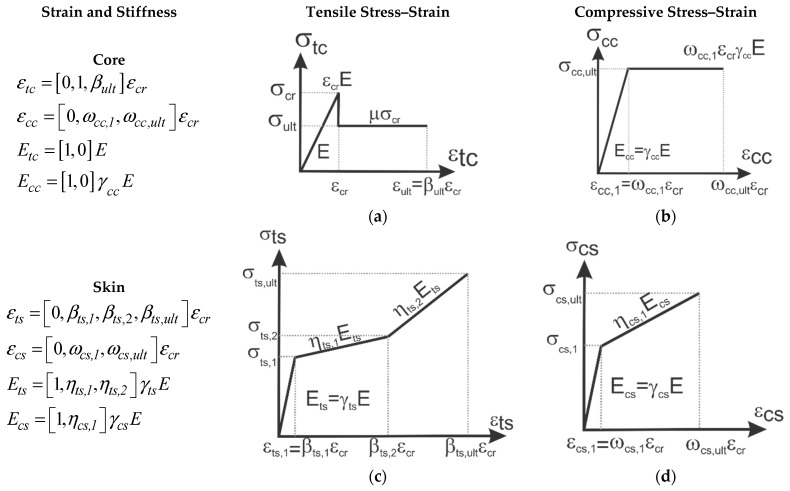
Constitutive models (**a**) and (**b**) elastic–perfectly plastic in tension and compression core (**c**) and (**d**) bilinear tension and elastic–perfectly plastic compression skin model.

**Figure 12 materials-14-00390-f012:**
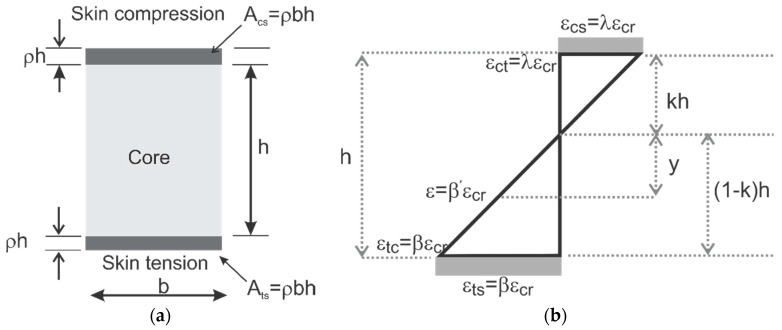
(**a**) Sandwich section, (**b**) linear strain distribution along the section.

**Figure 13 materials-14-00390-f013:**
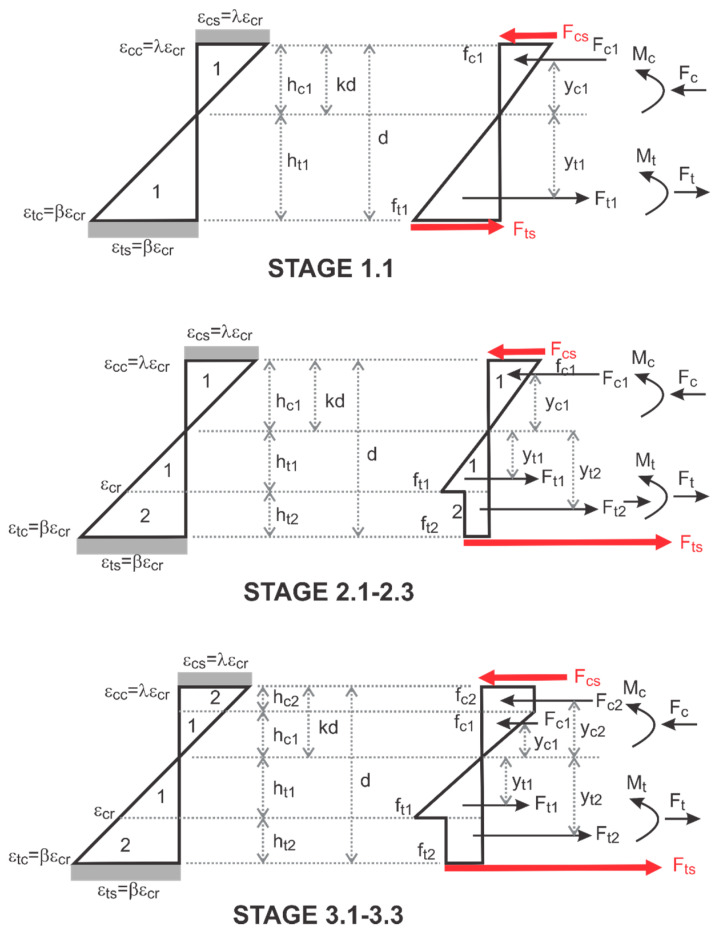
Strain–strain block at different stages with uniform stress in the skin.

**Figure 14 materials-14-00390-f014:**
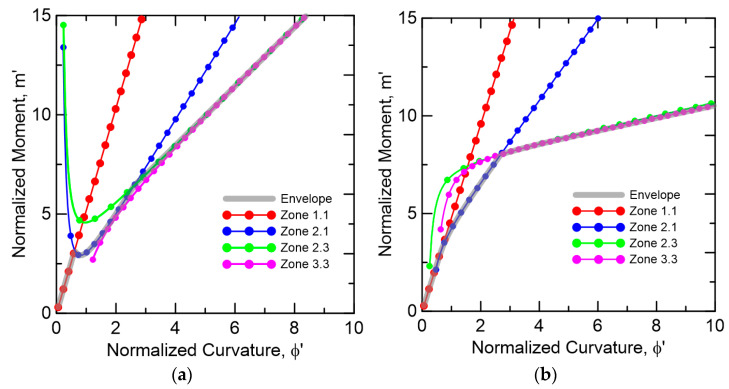
Moment–curvature response at each range of strain *β* (**a**) TRC-AAC (**b**) TRC-FRAC.

**Figure 15 materials-14-00390-f015:**
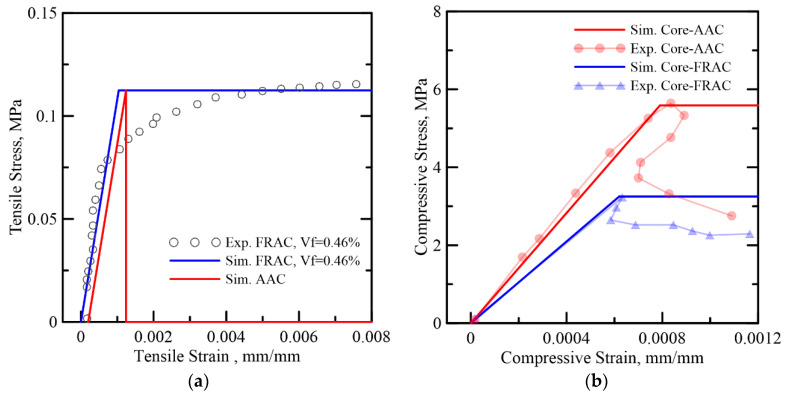
Simplified stress–strain model of AAC and FRAC core materials (**a**) tensile model and (**b**) compressive model.

**Figure 16 materials-14-00390-f016:**
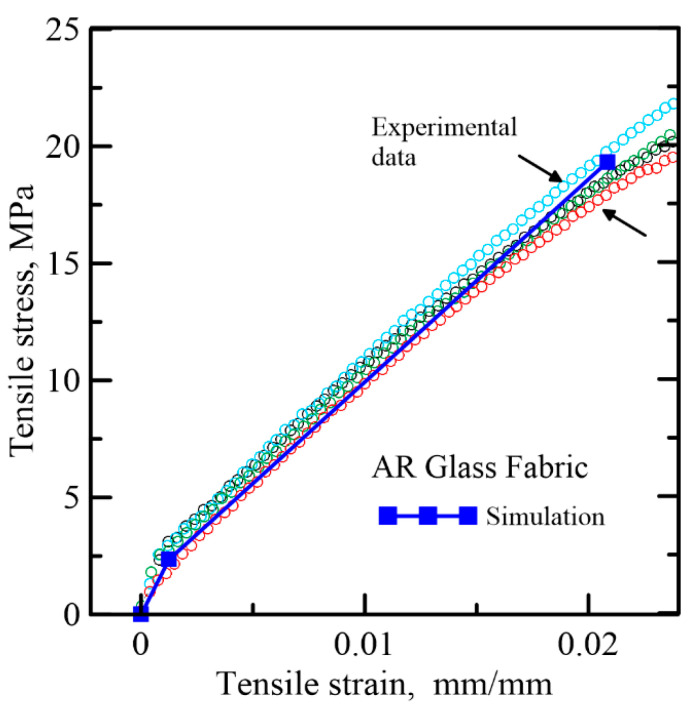
Simplify stress–strain diagram of skin materials, AAC and FRAC. [14].

**Figure 17 materials-14-00390-f017:**
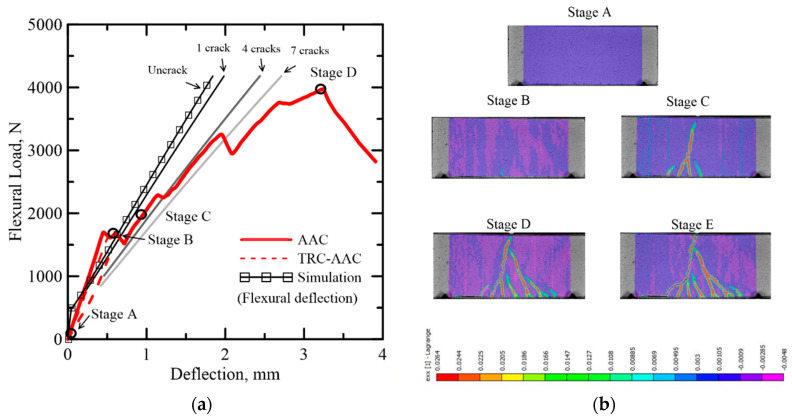
(**a**) Effect of number of cracks on the simulation of load-displacement response of TRC-AAC (50 × 100 × 200 mm), (**b**) DIC analysis on nominal longitudinal strain at various stages.

**Figure 18 materials-14-00390-f018:**
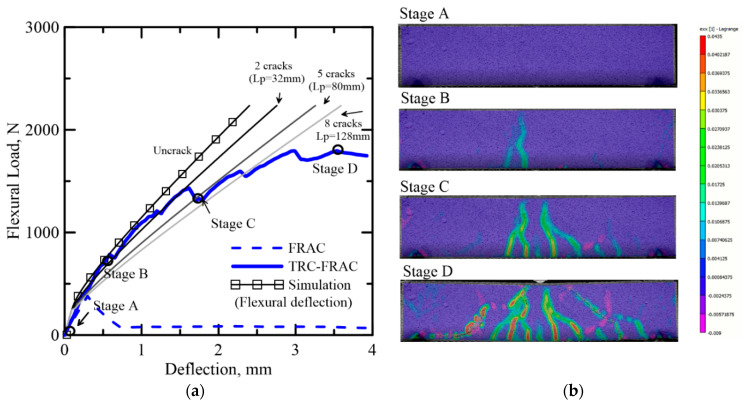
(**a**) Effect of number of cracks on the simulation of load-displacement response of TRC-FRAC (50 × 100 × 200 mm), (**b**) DIC analysis on nominal longitudinal strain at various stages.

**Figure 19 materials-14-00390-f019:**
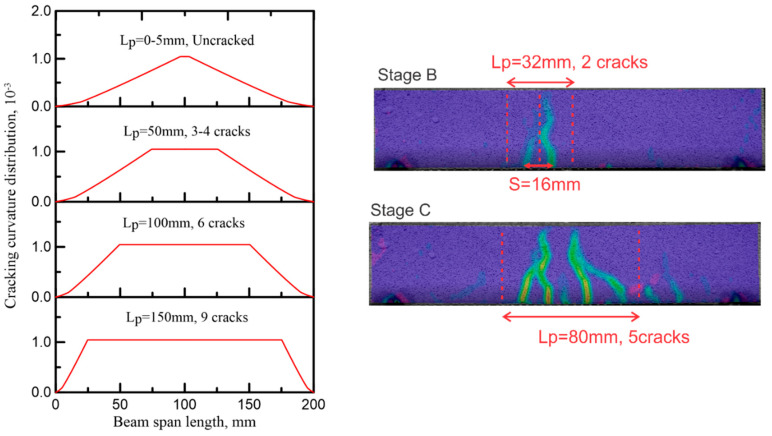
Correlation of simulation of average curvature at a function of number of cracks TRC-FRAC with dimension 50 × 100 × 200 mm.

**Figure 20 materials-14-00390-f020:**
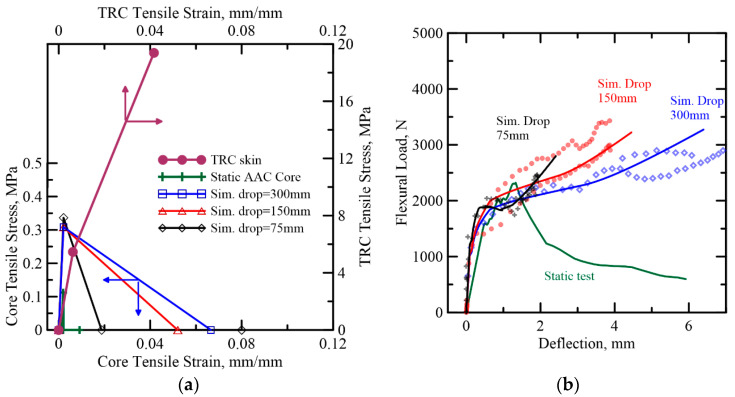
(**a**) AAC (B series) tensile strain–strain diagram from back-calculation of flexural impact tests (**b**) Comparison of the proposed model with the experimental data for TRC-AAC flexural impact.

**Figure 21 materials-14-00390-f021:**
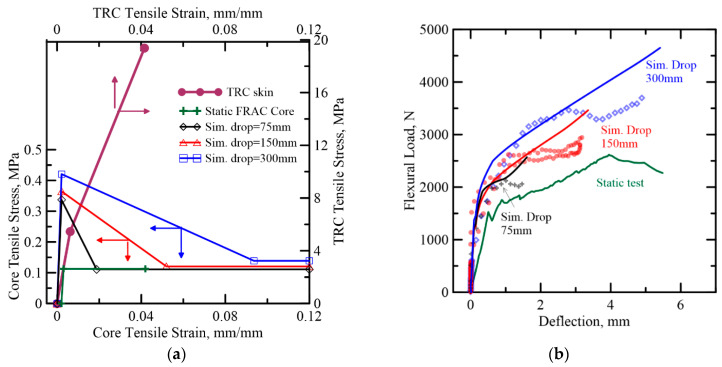
(**a**) FRAC (B series) tensile strain–strain diagram from back-calculation of flexural impact tests (**b**) Comparison of the proposed model with the experimental data for TRC-FRAC flexural impact.

**Table 1 materials-14-00390-t001:** Sandwich composite groups. (A) and (B) refer to specimen size designation

Designation	Skin	Core Material	Dimensions, mm
TRC-FRAC	2-ARG	FRAC	(A) 50 × 50 × 250 (B) 50 × 100× 250
TRC-AAC	2-ARG	AAC	(A) 50 × 50 × 250 (B) 50 × 100 × 250

**Table 2 materials-14-00390-t002:** Results of flexural impact tests on sandwich composite beams.

ID	Drop Height	Potential Energy [39]	Strain Rate [39]	Max Impact Force	Flexural Strength	Deflection at Max Force	Max Deflection	Toughness
	mm	J	sec^−1^	N	MPa	mm	mm	J
TRC-AAC-A	75	10.4	5	1690	7.2	1.52	12.8	8.77
	150	20.9	1	2117	9.3	3.96	14.42	25.82
TRC-FRAC-A	75	10.4	0.6	1946	8.1	1.12	9.89	9.75
	150	20.9	18	1653	6.7	0.77	19.3	16.95
TRC-AAC-B	75	10.4	0.1	2363	2.5	1.88	1.90	3.22
	150	20.9	0.7	3213	3.7	3.86	4.05	8.26
	300	41.8	3.1	3112	3.5	5.20	7.85	19.3
TRC-FRAC-B	75	10.4	0.2	2137	2.3	1.03	2.52	4.51
	150	20.9	0.5	2886	3.2	3.15	3.27	6.87
	300	41.8	3.0	3442	3.9	6.05	8.02	20.6

**Table 3 materials-14-00390-t003:** A normalized parameters for simulation of tensile and compressive response. b normalized parameters for simulation of tensile and compressive response of Skin for ARG-TRC.

a														
Core		*E* MPa		*ε_cr_ με*		*μ*			*β* *_ult_*		*γ* *_cc_*		*ω* *_cc,*1*_*	
AAC		108		1040		0			30		64		0.8	
FRAC		108		1040		1			30		48		0.6	
**b**														
***γ*** ***_ts_***	***β_ts,*1*_***		***β_ts,*2*_***		***β_ts,ult_***		***η*** ***_ts,*1*_***	***η*** ***_ts,*2*_***		***γ*** ***_cs_***		***ω*** ***_cs,*1*_***		***η_cs,*1*_***
17	1.2		1.2		24		1	0.46		17		1.2		0.46
*E_ts_* *MPa*	*ε* *_ts,*1*_* *με*		*ε* *_ts,*2*_* *με*		*ε* *_ts,ult_ με*		*E_ts,1_* MPa	*E_ts,*2*_* MPa		*E_cs_* MPa		*Ε_cs,*1*_* *με*		*E_cs,*1*_* MPa
1885	1264		1264		25,000		1885	868		1885		1264		868

**Table 4 materials-14-00390-t004:** Tensile parameters for impact testing.

Ref. Data	Tensile Parameters				Maximum Tensile Strength (MPa)	Tensile Toughness (MPa × 10^−3^)
	E (MPa)	*ε_cr_ (* *με* *)*	*μ*	*β* *_tu_*		
TRC-AAC						
Drop height 75 mm	162	2080	0	24	0.34	3.2
Drop height 150 mm	148	2080	0	43	0.31	8.0
Drop height 300 mm	148	2080	0	63	0.31	10.0
TRC-FRAC						
Drop height 75 mm	162	2080	0.33	30	0.33	15.8
Drop height 150 mm	175	2080	0.33	40	0.36	20.7
Drop height 300 mm	202	2080	0.33	50	0.42	29.8

## Data Availability

The experimental data of sandwich panels presented in this study are available in [Dey, V., Zani, G., Colombo, M., et al. “Flexural impact response of textile-reinforced aerated concrete sandwich panels,” *Materials and Design*, V. 86, 2015, pp. 187–97

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
