# Peer review of "Low-Velocity Impact Experiments and Modeling of TRC Skin-Aerated Concrete Core Sandwich Composites"

_materials, 2021, doi:10.3390/ma14020390_

Round 1
Reviewer 1 Report
In this study effect of a low-velocity impact is studied on TRC Skin-Aerated Concrete Core Sandwich Composites. The paper is overall well written and organized but the literature review is quite limited. Especially only two papers are cited from the last two years. There are several relevant papers published recently for different lightweight composites which should be referred to here (e.g., doi.org/10.3390/ma12183050, etc.) Other major observations are as follows,
1. Line-19 Stress skin was AR-glass. AR needs to be explained here either it is alkali-resistant glass or something else.
2. Line-119, Paragraph (Sample Preparation). FRAC and AAC static mechanical properties are obtained from ACI 523.4R-09. But please specify what is the mix recipe of both the core materials that are used to make sandwich panels for low velocity impact loading and flexural testing.
3. Line-198 Figure 3, Load vs deflection is divided into three zones linear elastic, increase deformation or yielding, and post-peak zone. But the flexural response of TRC-FRAC-50 x 100 is plotted till yield. It is suggested to include the post-yield response. Also, there is a need for quantification and comparison of the toughness of sandwich panels with the brittle and ductile core under static loading. The logic behind only considering the flexural response of samples of Category-B needs to be explained.
4. Line-220, What is the justification behind using 150 mm dropping height for impact testing? The justification needs to be properly elaborated with proper references.
5. Line-403, To validate the numerical model, simulation is performed for FRC-FRAC and FRC-AAC. But the validation of FRC-FRAC is done only at one thickness and the same is the case for FRC-AAC, validation for other parameters should be done as well. Moreover, the flexural results should also be validated as well.
6. Line-426, Figure 16 (a), the legends seem to be wrong here. please clarify.
7. Figures 19 & 20 captions are quite confusing. Please elaborate on the axis details.
Author Response
Low-Velocity Impact Experiments and Modeling of TRC Skin-Aerated Concrete Core Sandwich Composites
Chidchanok Pleesudjai, Anling Li, Vikram Dey, and Barzin Mobasher
Response to Reviewer Comments
The authors would like to acknowledge and thank the reviewers for thorough review and insightful comments. The reviewer comments have been addressed carefully and the manuscript has been revised accordingly. Response to the individual comments have been listed below in red.
Reviewer 1
General: In this study effect of a low-velocity impact is studied on TRC Skin-Aerated Concrete Core Sandwich Composites. The paper is overall well written and organized, but the literature review is quite limited. Especially only two papers are cited from the last two years. There are several relevant papers published recently for different lightweight composites which should be referred to here (e.g., doi.org/10.3390/ma12183050, etc.)
Response: The authors would like thank the reviewer for the comment. The introduction section of this paper has now been updated to include literature review based on the recent publications in this topic. Unfortunately, since the topic deals with aerated concrete and textile reinforced concrete, it was not possible to extend the models or tests to include lightweight concrete as well.
- Line-19 Stress skin was AR-glass. AR needs to be explained here either it is alkali-resistant glass or something else.
Response: AR-glass was changed to alkali-resistant glass (ARG) in the Abstract.
Line-119, Paragraph (Sample Preparation). FRAC and AAC static mechanical properties are obtained from ACI 523.4R-09. But please specify what is the mix recipe of both the core materials that are used to make sandwich panels for low velocity impact loading and flexural testing.
Response : An extra paragraph in Section 2.1: Specimen Preparation was added to summarize the mix designs for both FRAC and AAC materials in the revised version. Details on the specimen preparation and testing information is covered in a previous paper by Bonakdar et.al. The paper is referenced as:
Bonakdar A., Mobasher B., and Babbit F.,(2013) "Physical and mechanical characterization of fiber-reinforced Aerated Concrete (FRAC)", Cement & Concrete Composites, V. 38, pp. 82-9
- Line-198 Figure 3, Load vs deflection is divided into three zones linear elastic, increase deformation or yielding, and post-peak zone. But the flexural response of TRC-FRAC-50 x 100 is plotted till yield. It is suggested to include the post-yield response. Also, there is a need for quantification and comparison of the toughness of sandwich panels with the brittle and ductile core under static loading. The logic behind only considering the flexural response of samples of Category-B needs to be explained.
Response : The post-yield response of TRC-FRAC 50x100 has been added in Figure 4(d) (formerly Figure 3) in the updated version of this figure.
The quantification of toughness was also added in Section 3.1 comparing plain AAC and FRAC with TRC sandwich composites. Since the static mechanical tests were performed in a previous study by Dey et.al., we referenced that paper in our discussion, and detailed results from that study are not repeated in the current work. The paper is referenced as:
Dey V., Zani G., Colombo M., Di Prisco, M., Mobasher B.,(2015) “Flexural impact response of textile-reinforced aerated concrete sandwich panels”, Materials & Design, Volume 86, pp. 187-197.
In the updated paper, Figure 4(a) and 4(b) have been added to discuss the results from the Specimen Category- A i.e. core and sandwich specimen with FRAC and AAC with cross-section of 50 mmx 50 mm.
- Line-220, What is the justification behind using 150 mm dropping height for impact testing? The justification needs to be properly elaborated with proper references.
Response : Drop heights for the impact study were chosen based on the preliminary test runs. Core specimens, AAC and FRAC being light-weight and less stiff compared to other concrete materials offer less impact resistance and hence were tested with a drop height of max 150 mm. In order to establish direct correlation between the sandwich and plain aerated concrete core specimens, both materials were tested under identical drop heights. This facilitates determination of the absorbed energy capacity or toughness of different systems and comparison with the input potential energy from the impact event, later being directly related to the drop height and mass of the impact assembly (U= mgh) [12].
Dey V., Bonakdar A., Mobasher B.,(2014) “Low-velocity flexural impact response of fiber-reinforced aerated concrete.” Cem. Concr. Compos, 49, 100–110.
- Line-403, To validate the numerical model, simulation is performed for FRC-FRAC and FRC-AAC. But the validation of FRC-FRAC is done only at one thickness and the same is the case for FRC-AAC, validation for other parameters should be done as well. Moreover, the flexural results should also be validated as well.
Response: the authors appreciate reviewer's concern. Simulation of both 50 and 100 mm thickness specimens were performed during the study. In order to maintain the length of the paper at a reasonable level, authors decided to show TRC-AAC with t=100 mm and TRC-FRAC with t=50 mm to validate that the proposed method is capable of simulating both variables of core and thickness. In addition, it was necessary to dedicate a portion of the paper to the modeling and simulation since the analytical approach needed to be validated for other variables in the study. For example the number of parallel cracks and other variables of testing method such as range f strain rate, and mode of loading in impact/static testing are important variables.
The Figures show the simulation of TRC-FRAC and TRC-AAC in both t=50 and t=100mm. Note that in the absence of any failure criteria for the shear cracking in the core, the simulation of loading increases to much higher loads, therefore justifying for the need for proper incorporation of shear failure criteria.
- Line-426, Figure 16 (a), the legends seem to be wrong here. please clarify.
Response : Legend in figure 17(a) (formerly Figure 16(a)) has been updated to solid line
Figures 19 & 20 captions are quite confusing. Please elaborate on the axis details.
Response : Captions corresponding to Figures 20 and 21 (formerly Figure 19 and 20) have been corrected.

Reviewer 2 Report
Please see attached.

Author Response
Low-Velocity Impact Experiments and Modeling of TRC Skin-Aerated Concrete Core Sandwich Composites
Chidchanok Pleesudjai, Anling Li, Vikram Dey, and Barzin Mobasher
Response to Reviewer Comments
The authors would like to acknowledge and thank the reviewers for thorough review and insightful comments. The reviewer comments have been addressed carefully and the manuscript has been revised accordingly. Response to the individual comments have been listed below in red.
Reviewer 2
MAJOR Comment A. Flexural Modeling of Sandwich Response There are quite a few inconsistencies regarding the naming of the variables in the model incorporated by the authors.
Response : The parameters names were updated ref to Figure 10 of the updated paper. The nomenclature followed is described here. The first subscript of each parameter corresponds to the type of the stress (tensile (t), or compressive (c)), the second subscript to the domain (core (c), or skin (s)), and the third subscript to the stress/strain regime (1,2 or ultimate(ult)). However, the corresponding parameters for tension in core was kept the same as (). All subscripts of the variables were made as italic.
Comment B. Figure 10 The panels have different sizes and for 3 Some specific comments for each panel Panel a o s ult is not shown. Please add it on the left of the ordinate as well as a dotted line o Instead of me crE it would more clear to depict it as ms cr since it represents a non-dimensional stress; actually this is the definition presented in the manuscript (see line 307). o The stress in the ordinate is shown as s tc . Shouldn’t t e in the abscissa be denoted as tc e (tensile core), as well?
- Panel b, s cu is not shown. Please add it on the left of the ordinate as well as a dotted line. o
- Panel c, The right hand side of the panel is missing o Add a dotted line for s ts_ult as you did for ts1 ts2 s and σ
- Panel d, The right hand side of the panel is missing MINOR Comment C: Syntax/grammar corrections, acronyms and clarifications
Response : Parameters in Panels A to D have been updated corresponding to the revised parameters in comment A. Stress with subscript for cracking, post cracking and ultimate are added on the right hand side of the axis.
MINOR Comment C: Syntax/grammar corrections, acronyms and clarifications
Lines 6,10: “School of Sustainable Engng” >>
Response : This has been fixed by substitute “ Engng” with “Engineering”.
Line 46: “..400-800 kg/m3”
Response : This has been fixed, “kg/m3” has been replaced by “kg/m3”.
Line 50: “..Concrete (AAC). The..”
Response : The sentence has been modified, by capitalizing the first letter of the word “the”.
Line 79: “..consumption. also, the composites..”
Response : The sentence has been modified, by capitalizing the first letter of the word “also”.
Lines 124-125: “while the toughness of FRAC is more than 60 times AAC due to 124 the short polymeric fibers” 4 Improve English Table 1 Caption: Italize “F’c”
Response : The original sentence was modified. “The toughness of FRAC at displacement of 15 mm [7] was reported to be 19 Nm. This value is due to the addition of short polymeric fibers to FRAC and represents more than 60 times the toughness of AAC.” ”.
Lines 137-138: “solids. compressive strength at 28 days was 21 MPa (± 1.9 MPa) based on cylinder tests with a 137 nominal diameter of 50 mm and height of 100 mm.” The sentence starts abruptly, improve English.
Response : The sentence has been modified.
Line 139: “equal portions of freshly mixed cement paste is placed” “are placed”
Response : The sentence has been modified.
Figure 4: The legend of Fig. 4 for the continues lines reads “Exp.” which corresponds to LVDT is I understand correctly. If that’s so, please replace “Exp.” with “LVDT” in the legend.
Response : Correct, The legend of Figure 4 was updated as per comment.
Lines 239-240: “the first crack forms in point (A) Figure 239 6a.” Improve English.
Response : The sentence was replaced by “Stage I is the linear elastic range up to the limit of proportionality (LOP) representing by point A in Figure 7(a) (formerly Figure 6(a)).”
Line 239: I cannot seem to spot the failure mechanism corresponding to the roman numeral IIX (8?!?) in Fig. 9. It should correspond to the panel VIII (8). IIX is not valid, I think, thus change it to VIII.
Response : Agreed, this was a typographical error. The graphic callout of Figure 9 has now been modified based on the comment.
In lines 306-308 the article reads: “In the tensile domain, after cracking a constant residual strength defined by s ms ult cr = is used, with m
Response : A constant residual strength was replaced to “non-dimensional residual strength” per this comment.
Comment D. Fig. 6b. The colors in Fig. 6b depict some kind of strain (Cauchy, engineering?). Please add an explanation in the caption of the figure.
Response : The DIC analysis in this paper had been used VIC-3D program which calculate the Lagrangian strain tensor on the specimen surface(https://www.correlatedsolutions.com/digital-image-correlation)
Lagrangian strain = ½( λ2 − 1 ) where λ is the stretching element/original element. Title of Figure 7 (formerly Figure 6) has been modified to add reference to this Lagrangian strain tensor measurement by DIC technique.
Comment E. Fix the Figure hyperlinks Some of the Fig. hyperlinks have been broken. See lines: 237, Fig. 6; 254, Fig. 7; 267, Fig. 8. This might have happened because you changed the letters for these figures manually instead of refreshing the document. A way to make sure there aren’t any broken hyperlinks is to either refresh the document (ctrl+shift+F11 (in case the hyperlinks are locked) → ctrl+A → F9 or to simply save the document as .pdf, since during saving it performs an automatic refresh to the document
Response : Thank you for suggestion
Reviewer 3 Report
The reviewed paper deals with experiments on two kinds of sandwich aerated concrete composites. Two kinds of core and two types of skins have been considered. The research is interesting can fits to the journal scope. Unfortunately, in the current version several drawbacks can be indicated.
Here only general problems will be discussed. Other are indicated in the attached reviewed version of the paper.
- The experimental paper should be constructed in such a way that the other research center (having access to similar equipment) can repeat the experiments to verify them. In the current paper several data are not given (details indicated in the paper).
- The Authors used two ways of results registration during experiments. Unfortunately, they have got significant differences in the deflection results. They describe potential reasons of differences instead of improvement of experiments to check which results are correct. The defection date differs qualitatively but also quantitatively, what has influence on identification results. That means that presented in Chapter 4 analysis using these data are questionable.
- Chapter 4 is devoted to modeling of sandwich response. It should be based on experimental results from previous sections of the current paper. Chapters 4.3 and 4.4 present identification based on previous experiments (from [11] and [17]) and are out of the main paper subject. Of course the results of previous Authors’ research can be used in the current research, but it should be presented as the final result (e.g. in tables).
- The number of self-citations is very high (more than 25%).
- The paper has no significant conclusions.

Author Response
Low-Velocity Impact Experiments and Modeling of TRC Skin-Aerated Concrete Core Sandwich Composites
Chidchanok Pleesudjai, Anling Li, Vikram Dey, and Barzin Mobasher
Response to Reviewer Comments
The authors would like to acknowledge and thank the reviewers for thorough review and insightful comments. The reviewer comments have been addressed carefully and the manuscript has been revised accordingly. Response to the individual comments have been listed below in red.
Reviewer 3
The reviewed paper deals with experiments on two kinds of sandwich aerated concrete composites. Two kinds of core and two types of skins have been considered. The research is interesting can fits to the journal scope. Unfortunately, in the current version several drawbacks can be indicated.
Here only general problems will be discussed. Other are indicated in the attached reviewed version of the paper.
- The experimental paper should be constructed in such a way that the other research center (having access to similar equipment) can repeat the experiments to verify them. In the current paper several data are not given (details indicated in the paper).
Response : The authors agree with the reviewer comment. Some discussion on the manufacturing process, mix design of core materials has now been added in the Section 2.1 of the updated paper. Several previous papers have been referenced in the discussion since the test method development and experimental facilities and testing and data reduction procedures have all been developed in the past 10 years and published elsewhere. We have also included a new Figure 1, which summarizes the key material properties of AAC and FRAC, compiled from previous studies. Composition of the TRC skin layer used in the sandwich composite along with the specimen sizes has also been discussed in the same section 2.1. Description of the static and flexural impact test setup and DIC technique used in this study has been presented in the Section 2.2 and 2.3, respectively. Detailed response to the individual comments in the paper is presented below:
- Section 2.1 Sample preparation
Static mechanical test of core materials (AAC and FRAC) using in sandwich composite reference to the mechanical testing paper in [7]. The full details are found in [7]. However, some discussion on the have been added in this section, see below:
“AAC specimen was manufactured by using 18% of Portland cement, 27% of silica, 8% of limestone combined with gypsum and 9% recycled coal combustion by-product. On the other hand, FRAC core was manufactured with 28% cement and 42% fly ash. An extra 4.6% fiber was added to only the FRAC core in order to activate the ductility. Water was used by 38% in AAC and 28% in FRAC. Note that percentages are presented by weight. Static mechanical test results of the two core types are listed in Figure 1 tested by closed-loop displacement control [7]”
- Figure 2(a), schematic figure may not fit the real dimension, but on the right-hand side,
- Figure 2(b), shows an actual specimen and fits the data in Table 2. There is no modification in this figure.
- Supporting span was added in paragraph (200mm).
- The hammer weight was added “The impact test set-up shown in Figure 2 is based on the free-fall of an instrumented hammer with the weight of 134N on a specimen under three-point bending with support span of 200mmThe hammer’s force is measured by strain-gage based load-cells of 90 kN capacity mounted behind the blunt shaped impact head.
Section 3.1 Static test
- Figure 3 were add the results of specimen 50x50mm and has been discussed with the result of 50x100mm
Section 3.4 Dynamic response
- The number in parentheses refers to English unit. It was eliminated in the revision document.
- The Authors used two ways of results registration during experiments. Unfortunately, they have got significant differences in the deflection results. They describe potential reasons of differences instead of improvement of experiments to check which results are correct. The defection date differs qualitatively but also quantitatively, what has influence on identification results. That means that presented in Chapter 4 analysis using these data are questionable.
Response: Because of the large deformation of the specimen during the impact event, the range of the LVDT used and its frequency response could introduce some errors in the signal. Using the DIC analysis technically is the favorable choice and more robust way of measuring the deflection response since DIC measured the relative deflection on the surface of the specimen and eliminates the spurious deformations The DIC measurements are also validated by other specimens tested on the same instrument and other experiments conducted at much higher strain rates. This paper points out that the robust data presented by DIC analysis conducted on the previously conducted tests in [17]. Therefore, load-deflection response by DIC also was in used in Section 4.
- Chapter 4 is devoted to modeling of sandwich response. It should be based on experimental results from previous sections of the current paper. Chapters 4.3 and 4.4 present identification based on previous experiments (from [11] and [17]) and are out of the main paper subject. Of course the results of previous Authors’ research can be used in the current research, but it should be presented as the final result (e.g. in tables).
Response : All simulations in Section 4, either static and impact are based on the same data presented in the present manuscript. All comparisons are made with the data analyzed and reported within the span of this manuscript. The sandwich composites presented in this paper were manufactured by the skin and core materials presented in the previous work by author [Mobasher, 2011, and Bonakdar et. al. 2013] Material properties were adopted to calibrate the material parameters for the proposed model. The reference callout [11] was incorrect in the original version of the paper, it was supposed to be [7] and has now been fixed in the revised document. The testing results of flexural response of sandwich composite was originally validated in previous paper [17]. At that time, the result presented in the case of load-displacement by LVDT. That data was re-analyzed in this paper to understand the difference between deflection measurment by DIC compared to LVDT.
- The number of self-citations is very high (more than 25%).
Response : References 7, 12, and 17 are self-citations used in the current work to reference author’s previous experimental work on Aerate Concrete . Whereas references 18 and 26 are the primary papers published by the corresponding author on this back-calculation approach. References 32, Hence we feel that these references are necessary and must be cited in the current work.
- The paper has no significant conclusions.
Response :
The conclusions section has been rewritten to highlight the major contributions of the work.
Conclusions
Flexural experiments on the response of textile-reinforced aerated concrete sandwich panels were conducted and the results modeled under quasi-static and low-velocity impact loads. Plain and fiber-reinforced aerated concrete were used as core and AR-glass textile concrete was used as the skin element. Specimens exhibit the characteristic response of ductile core/ductile skin compared with brittle core/ductile skin. A parametric model was proposed to simulate the contribution of the two phases. Closed-form moment-curvature relationship is obtained by the integration of section equilibrium, strain compatibility, and piecewise linear constitutive law. The interaction of the material properties in terms of strain capacity and stiffness of each phase can be addressed by the parametric model and used for the optimization of material properties. Materials parameters of individual core and skin in the proposed model have been calibrated based on direct compression and tension testing. Furthermore, the proposed model was used back-calculation process to assess tensile response in core with the variable of drop height. Static and impact test results are simulated using closed-form solutions. This approach allows parameter estimation as well as a standard procedure to analyze and design sandwich composites. The key experimental findings are summarized below:
- The additional TRC layers significantly increased the toughness by a factor of 22 and 18 in TRC-AAC 50x50 and TRC-AAC 50x100, respectively, compared to the plain AAC cores at deflection limit of 5 mm whereas toughness of TRC-FRAC 50x50 and TRC-FRAC 50x100 increased by a factor 12 and 2, respectively to plain FRAC cores.
- In high energy events, the shear failure mode dominates the sandwich response, with the diagonal tension cracks taking place at the ultimate strength point indicating that the strength of the core system is the detrimental failure mode. The tension TRC skin carries a significant amount of load and provides the strength and ductility to the composite as shown by the saturated lateral cracks and spalling regions as well as debonding between the textile and matrix.
- The dynamic flexural strength and toughness increase correlated to the dop height in range of 2-8 MPa and 4-21 J for TRC-FRAC whereas 2-9 MPA and 3-26 J for TRC-AAC.
- The simulation of flexural response by the proposed model is well correlated with the data measuring by DIC when the difference between the measurements in LVDT and DIC is due to three main parameters; a) frequency signals, b) spurious deformations correlated with the mechanical measurement of the displacement and c) Adjustable between relative vs. global displacement.
- The interaction of the material properties in terms of strain capacity and stiffness of each phase can be addressed by the parametric model and used for the optimization.
- Simulation of the flexural response is dependent on the length of localization and by changing this parameter the sequence of cracking can be matched against the DIC analysis.
- The shear effect and shear failure criteria are important aspects of the final failure mode in the core. Currently, the proposed model ignores the shear contribution to the failure but it can be predicted through a strength-based on diagonal tension mechanism.
Reviewer 4 Report
The article can be published.
I only found two minor things that should be changed.
Line 79. "also" should be replaced by "Also"
Lines 237, 254, 267. Remove - "Error! Reference source not found."
Author Response
Low-Velocity Impact Experiments and Modeling of TRC Skin-Aerated Concrete Core Sandwich Composites
Chidchanok Pleesudjai, Anling Li, Vikram Dey, and Barzin Mobasher
Response to Reviewer Comments
The authors would like to acknowledge and thank the reviewers for thorough review and insightful comments. The reviewer comments have been addressed carefully and the manuscript has been revised accordingly. Response to the individual comments have been listed below in red.
Reviewer 4
General: The article can be published.
I only found two minor things that should be changed.
Line 79. "also" should be replaced by "Also"
Response : “a” changed to “A”
Lines 237, 254, 267. Remove - "Error! Reference source not found."
Response : - The referencing error has been fixed.
Round 2
Reviewer 1 Report
The authors have done significant revisions. However, the introduction section is very weak in terms of literature review. There are several chunks of text, which need to be properly referenced.
For example,
Line 34-37
Line 39-46
Line 59-62
Line 70-73
Author Response
The authors have done significant revisions. However, the introduction section is very weak in terms of literature review. There are several chunks of text, which need to be properly referenced. For example, Line 34-37 Line 39-46 Line 59-62 Line 70-73
Response : The introduction part has been modified based on the comments. A total of 14 additional references that describe the recent work on lightweight concrete and AC have been added. Various features of TRC sandwich composites were also referenced by several up-to-date papers (within the past 5 years). The sections that lacked proper references were eliminated or modified with extra references.
Reviewer 2 Report
The authors addressed all of my comments. The overall quality and readability of the manuscript, and the ease to understand the underlying model has been improved substantially.
A minor note: in my reply I meant that the subscripts denoting the domains and regimes should be in Roman and not in italics; however, that's a very minor issue since the document appears to be consistent with this convention. In general, it is best to italize variables, retain the descriptors/indeces (e.g., denoting the domain) with Romans and present the vectors with bold fonts.
Author Response
The authors addressed all of my comments. The overall quality and readability of the manuscript, and the ease to understand the underlying model has been improved substantially. A minor note: in my reply I meant that the subscripts denoting the domains and regimes should be in Roman and not in italics; however, that's a very minor issue since the document appears to be consistent with this convention. In general, it is best to italize variables, retain the descriptors/indexes (e.g., denoting the domain) with Romans and present the vectors with bold fonts
Response : Thank you for your recommendation. There is no response to this comment.
Reviewer 3 Report
The paper has been considerably improved. The description od the experiments is much better. The conclusion are specified. The paper can accepted in the current form.
Author Response
The paper has been considerably improved. The description of the experiments is much better. The conclusion are specified. The paper can accepted in the current form.
Response : Thank you for your recommendation. There is no response to this comment.